# Insight into the Inhibitory Mechanism of Aryl Formyl Piperidine Derivatives on Monoacylglycerol Lipase through Molecular Dynamics Simulations

**DOI:** 10.3390/molecules27217512

**Published:** 2022-11-03

**Authors:** Chang Liu, Shanshan Guan, Jingwen E, Zhijie Yang, Xinyue Zhang, Jianan Ju, Song Wang, Hao Zhang

**Affiliations:** 1Laboratory of Theoretical and Computational Chemistry, Institute of Theoretical Chemistry, College of Chemistry, Jilin University, Changchun 130023, China; 2College of Biology and Food Engineering, Jilin Engineering Normal University, Changchun 130052, China; 3Key Laboratory of Molecular Nutrition at Universities of Jilin Province, Changchun 130052, China

**Keywords:** monoacylglycerol lipase, inhibitor, depression, molecular dynamics simulations, MM/PBSA

## Abstract

Monoacylglycerol lipase (MAGL) can regulate the endocannabinoid system and thus becomes a target of antidepressant drugs. In this paper, molecular docking and molecular dynamics simulations, combined with binding free energy calculation, were employed to investigate the inhibitory mechanism and binding modes of four aryl formyl piperidine derivative inhibitors with different 1-substituents to MAGL. The results showed that in the four systems, the main four regions where the enzyme bound to the inhibitor included around the head aromatic ring, the head carbonyl oxygen, the tail amide bond, and the tail benzene ring. The significant conformational changes in the more flexible lid domain of the enzyme were caused by 1-substituted group differences of inhibitors and resulted in different degrees of flipping in the tail of the inhibitor. The flipping led to a different direction of the tail amide bond and made a greater variation in its interaction with some of the charged residues in the enzyme, which further contributed to a different swing of the tail benzene ring. If the swing is large enough, it can weaken the binding strength of the head carbonyl oxygen to its nearby residues, and even the whole inhibitor with the enzyme so that the inhibition decreases.

## 1. Introduction

It is estimated that depression accounts for a greater proportion of the world’s disease burden than any other illness and has become one of the most common mental illnesses to afflict human beings [1]. However, its etiology has remained largely unexplained [2]. The idea that the endocannabinoid system (ECS) may be associated with emotional behavior, particularly depression, is rooted in the fact that cannabis consumption in humans has a profound effect on mood [3]. The investigation shows that ECS is involved in various physiopathological processes in the central and peripheral nervous systems and peripheral organs [4]. Regulating the activity of the ECS may have therapeutic potential for a wide range of human diseases such as neurodegenerative diseases [5], obesity/metabolic syndrome [6], inflammation [7], cardiovascular [8], liver [9], pain [10], cancer [11], and psychiatric disorders [12].

The ECS is comprised of endocannabinoid receptors, endocannabinoids, and enzymes that are responsible for the synthesis and degradation of endogenous cannabinoids [13]. Among the cannabinoids targeting endocannabinoid receptors, the lipid derivatives named Anandamide (AEA) and 2-Arachidonoyglycerol (2-AG) are mainly studied [14,15]. The enzymes that degrade the two substances both belong to the serine hydrolase superfamily. Fatty acid amide hydrolase (FAAH) is mainly responsible for the degradation of AEA, while 2-AG is mostly inactivated by hydrolysis of its ester bond through the presynaptic localization enzyme Monoacylglycerol lipase [16,17]. 2-AG is the most abundant endocannabinoid in the brain and is associated with the physiopathology of depression [18,19]. As a result, there is a hope that research on MAGL inhibitors will provide treatment for depression.

MAGL belongs structurally to the α/β hydrolase superfamily [20] and is a membrane-associated hydrolase. It contains a total of 303 residues (as shown in Figure 1), and the center of the fold consists of seven parallel and one antiparallel β-pleated sheets, which are surrounded by eight α helixes. The residues of Ala151-Pro225 in MAGL form the lid domain [21]. The α4 helix is an amphiphilic helix with the hydrophobic part facing the membrane which facilitates interaction with the cytomembrane for the recruitment of lipophilic substrates [22]. The structure is highly flexible to the extent that it exhibits two different conformations in the crystal: open [23] and closed [21]. The active site of MAGL is in the substrate binding pocket behind the lid domain, with the most important catalytic triads including Ser122, Asp239, and His269 [24]. During the hydrolysis of the catalytic ester bond, the carbonyl oxygen on the substrate ester bond can develop an oxygen anion and bind to the nearby oxyanion hole region [25] (Gly50, Ala51, Met123, and Gly124). Therefore, the oxyanion hole is also a region that has a strong influence on catalytic activity.

In recent years, there have been an increasing number of inhibitors of MAGL as the physiological functions of MAGL are constantly studied [26]. So far, several types of MAGL inhibitors have been reported, depending on how the enzyme interacts with the inhibitor. These inhibitors can be classified into irreversible inhibitors and reversible inhibitors [17]. It was found that permanent blockade of MAGL activity by irreversible inhibitors can produce pharmacological tolerance and functional antagonism involving similar forms of receptor desensitization [27]. Consequently, medicinal chemists have turned their attention to the discovery of reversible inhibitors. In 2009, Piomelli’s group identified the first reversible inhibitors of MAGL, which were natural terpenoids [28]. Subsequently, reversible inhibitors such as β-amyrin, benzodioxole, and piperazinyl pyrrolidin-2-ones were discovered [29,30,31]. Recently, treating MAGL as a potential therapeutic target for depression, Zhuoer Zhi et al. were the first to develop an aryl formyl piperidine derivative inhibitor that was found to significantly improve depressive-like behavior induced by reserpine [32]. We selected four more representative inhibitors as the main research objectives of this work (the chemical names of the four inhibitors are detailed in the abbreviations section). It is shown in Figure 2 that the parent nucleus structures and 4-substituted groups of the four inhibitors are identical, with only the 1-substituted groups differing, which caused an approaching 7000-fold and more differences in inhibitory capacity in the experiment. However, the mechanism of their inhibition at the molecular level has not been clarified.

For this study, molecular dynamics (MD) simulations were used to investigate the binding mode of MAGL to the four inhibitors, while molecular mechanics/Poisson–Boltzmann surface area (MM/PBSA) methods were used to calculate the binding free energy between enzymes and inhibitors to investigate the differences in the effects of different 1-substituted aryl formyl piperidine derivatives on the protein and to reveal the inhibitory mechanism of the inhibitors. This could provide some information on the development of antidepressant drugs.

## 2. Results and Discussion

### 2.1. Molecular Docking Results

We first verified the reliability of the docking results. In the X-ray crystal structure (PDB ID: 3PE6), MAGL was bound to (2-cyclohexyl-1,3-benzoxazol-6-yl) {3-[4-(pyrimidin-2-yl) piperazin-1-yl] azetidin-1-yl} methanone inhibitor (abbreviated as ZYH). It was removed and redocked back to the protein, and the complex was obtained by docking in which the spatial orientation of ZYH was essentially the same as that in the crystal structure. High-affinity energy (−9.2 kcal·mol^−1^) was observed between the redocked ZYH and MAGL, and the residues forming hydrogen bonds (Ala51 and Met123) were consistent within the crystal structure. The above work demonstrates that the docking findings were qualitatively consistent with the experiment.

After ensuring the reliability of the molecular docking simulation, we docked MAGL and inhibitors 1–4, respectively. The results of the docking are shown in Appendix A. The obtained complexes were named complexes 1–4, correspondingly. There were no significant differences in the positions and binding postures of the ligands in the four complexes. Except for complex 1, hydrogen bonds were formed in all three complexes. Inhibitor 2 and inhibitor 3 both formed two hydrogen bonds with Ala51 and Met123, respectively, while inhibitor 4 formed only one hydrogen bond with Met123.

### 2.2. Stability of Molecular Dynamics Simulations

Classical molecular dynamics simulations were simulated for 200 ns with the four protein-inhibitor complexes, considering the docking results as the initial conformation. As a reference, we also performed a 200 ns simulation for the independent protein of MAGL. The root-mean-square deviation (RMSD), as a measure of the stability of a molecular system, represents the deviation of the simulated structure from the initial structure (reference structure) at a certain moment in time. The RMSD of the Cα atoms for the four complexes and independent protein throughout the simulation is shown in Figure 3. As can be seen from the figure, all five systems were balanced after 20 ns. The fluctuations in RMSD were less than 0.1 nm, indicating that the system had stabilized.

### 2.3. Root-Mean-Square Fluctuation and Flexibility Analysis

The root-mean-square fluctuation (RMSF) describes the flexibility of regions in a molecular system. When a ligand binds to a region of a protein, the residues in the region are limited by the ligand. This will result in reduced flexibility and lead to a lower RMSF. Figure 4a–d shows the RMSF of independent protein in comparison to complex 1, complex 2, complex 3, and complex 4, respectively. The position of the U-shaped structure in the protein is shown in Figure 4e. The α4 helix and the loop connecting α4 and α5 helix constituted a U-shaped structure together (Ala151–Arg186). The RMSF values of the U-shaped structure for the four complexes became smaller. The RMSF peak values (Asn168) of the independent protein to the four complexes were reduced from 0.4273 nm to 0.2290, 0.2973, 0.2683, and 0.3399 nm, respectively. In the four complexes, there was also some reduction of RMSF values in the loop_237–242_ and loop_265–269_ regions (where Asp239 and His269 were in the catalytic triads). It is suggested that the inhibitor occupies the key catalytic region and there may be some interactions with the main catalytic residues, which make it less likely to leave the active site, exhibiting the characteristics of a competitive inhibitor as mentioned in the literature. In addition, the RMSF values of the Asn187-Leu199 region (mainly the α5 helix) were significantly decreased in complex 2 and complex 3 compared to independent protein, whereas the decrease was not significant in complex 1 and complex 4. It indicates that residues of the region in complex 2 and complex 3 may be stronger interacting with the inhibitor.

### 2.4. Principal Component Analysis (PCA)

The principal component analysis was performed on the four complexes in order to count the relative variations in regions of the protein molecule system during the simulation. The percentages of the first five eigenvectors for all systems are shown in Appendix A, and the total percentages of the first five eigenvectors in each system reach more than half. The first two eigenvectors were chosen to build the covariance matrix and analyze the trend of mutual motion between the residues. Figure 5 shows the covariance matrixes for the four complex systems, where larger absolute values indicate a higher correlation between the variables. It is clear from the observation in Figure 5 that a segment of the U-shaped structure (Ala151–Ser176) in each of the four complexes has a significant negative correlated motion relative to most of the other regions of the protein (as shown in the green rectangles in Figure 5). It demonstrates that there is an obvious relative motion between the U-shaped structure and the rest of the protein during the simulation. Moreover, there was a remarkable positive correlated motion of the Asp76–Asp92 region relative to the region with a smaller residue number, while there was likewise a remarkable positive correlated motion of the Thr260–Leu275 region relative to the Pro230–Arg240 region (as shown in red rectangles in Figure 5, respectively).

### 2.5. Cluster Analysis, Free Energy Landscape (FEL), and Sampling

Clustering can identify representative conformations in the sampling set [33], which enables us to acquire samples with lower energy and in the largest cluster to serve as subjects for subsequent conformational analysis. The proportion of clusters with different conformations in the four complexes is shown in Appendix A, where the proportion of the largest clusters in the different complexes is 55.7% (complex 1), 54.4% (complex 2), 68.5% (complex 3), and 47.5% (complex 4), respectively. The clustering conformations of the largest clusters in complex 1, complex 2, and complex 3 were more than half, with only complex 4 being slightly less than half.

The free energy landscape (FEL) illustrates the conformational energy changes in the MD simulation trajectory [34]. Figure 6 portrays the FEL obtained from the projection of the first two eigenvectors (PC1 and PC2) in the four complexes, which reflects the distribution of conformational energy. The darker the red color, the lower the energy of the system. In combination with the results of the cluster analysis, two snapshots were selected for sampling in each of the four molecular systems that were relatively low in energy and accounted for the largest proportions of the respective clusters. One of these samples served as the primary target for subsequent conformational observations (the conformation of complex 1 at 90.74 ns, the conformation of complex 2 at 123.13 ns, the conformation of complex 3 at 97.82 ns, and the conformation of complex 4 at 87.30 ns, whose 3D structures were placed in the Appendix A), but the results observed in one sample were confirmed in the other.

### 2.6. Conformational Analysis of Samples

Comparing the sample conformations of the four complexes (as shown in Figure 7a), it was found that the conformations of the lid domain differed considerably in the different complexes. For the four complexes, the average values of RMSD in the lid domain were 0.1934, 0.1970, 0.1775, and 0.2002 nm, respectively, while the average values of RMSD for the whole protein were 0.1465, 0.1442, 0.1270, and 0.1463 nm, respectively, showing that the changes in the lid domain during the simulation were significantly higher than the whole protein. As mentioned in the introduction, there were three larger helixes in the lid domain, of which the head of the inhibitor (1-substituted terminal) was primarily bound to the α4 and α6 helixes, and the tail (4-substituted terminal) was primarily bound to the α5 helix. There is another comparatively short helix (Asp180–Leu184) on the tail side as well. The variability of the four helixes in the sampled conformations is shown in Figure 7b–e. The α4 helix displays the greatest conformational change. The helix_180–184_, the α5 helix, and the α6 helix also exhibit some conformational changes within the four complexes. The conformations of the α5 helix and helix_180–184_ exhibited some conformational differences due to the position of the inhibitor tail. The conformational changes observed in the samples could be confirmed by the statistical evidence from RMSD. The average values of RMSD for the four helixes are listed in Table 1, and from the table, we found that the average values of RMSD for the α4 helix were significantly higher than the other three in the four complexes. It was worth noting that the a4 helix of complex 1 showed partial unwinding in the sample conformation. The unwinding of this helix would also be analyzed and discussed for the total snapshots later in the secondary structure study. Additionally, the loop (part of the U-shaped structure) connecting α4 helix and helix_180–184_ showed a larger conformational change. The average values of RMSD for this loop in complexes 1–4 were 0.1002, 0.1012, 0.1008, and 0.1107 nm sequentially.

The conformations of the inhibitors in the different complexes differed somewhat as well (as shown in the enlarged frame in Figure 7a). The position of the carbonyl oxygen between the aromatic ring of the head and the parent nucleus overlapped quite well. This carbonyl oxygen was closer to the oxyanion hole in the protein, which probably had strong interactions. The tails of inhibitors displayed visibly positional changes in the different complexes. On closer observation, this change in position was shown to be the result of the combined flipping and lateral swing of the tail group. This flipping was evidenced by the direction of the carbonyl oxygen on the tail amide bond and the Cl atom on the tail benzene ring. It was particularly in complex 3 and complex 4 that the tail benzene ring was flipped by almost 180° compared to complex 1 and complex 2 (as shown in Figure 8). As a comprehensive result, the largest swing occurred in the tail benzene ring of inhibitor 4. With reference to helix_180–184_ on the lid domain, the distances between the tail benzene rings of the inhibitors and this helix in the four complexes were decreased in the order of 2, 1, 3, and 4 (the closer to this helix, the more deviation from the initial position).

We used the default parameters of PyMOL to search for possible hydrogen bonds in the sample conformations. Ala51 and Met123 were all observed to form hydrogen bonds with the inhibitor in the four complexes. The formation of hydrogen bonds with the inhibitor was observed for Ser122 in complex 1 and Glu53 in complex 2 as well. Additionally, a few residues with large hydrophobic side chains that were in proximity to the inhibitor may have hydrophobic interactions with the inhibitor. Ile179, Tyr194, Leu241, and Val270 were present in all complexes. The positions of the above residues that may form interactions are shown in Figure 9.

### 2.7. Hydrogen Bonding Analysis

Observing some of the hydrogen bonds in the sample of Figure 8, a hydrogen bonding analysis was then carried out to confirm the prevalence of these bonds throughout the simulation. The average number of hydrogen bonds among all snapshots in complexes 1–4 was 2.4110, 2.0441, 1.9524, and 0.9087 in that order, which was consistent with the IC_50_ values of the inhibitors measured in the literature [32]. The inhibitors studied in this paper were competitive inhibitors, and their inhibitory capacity was related to enzyme–inhibitor affinity. As the inhibitor binding sites were similar, there was little difference in their non-directional hydrophobic interactions. The main differences in affinity between the different inhibitors were probably more in the directional hydrogen bonding and electrostatic interactions. Thus, the number of hydrogen bonds was correlated to some extent with the inhibitory capacity of the inhibitor. The average number of hydrogen bonds in complex 4 was significantly lower than the other three complexes, which could be attributed to the largest swing in the tail of complex 4, affecting relatively stable hydrogen bonds that were originally existing. Furthermore, the occurrence percentages of hydrogen bonds between different residues and inhibitors were calculated for all snapshots (as shown in Table 2). In the four complexes, both Ala51 and Met123 near the oxyanion hole formed more hydrogen bonds with the carbonyl oxygen atom at the head of the inhibitor. The occurrence percentages of the two hydrogen bonds in complex 4 were significantly reduced, suggesting that the larger swing in the tail of complex 4 affected the stability of the oxygen anion hole bound to carbonyl oxygen to some extent. In addition, Ser122 in complex 1 also formed hydrogen bonds with the inhibitor tail. Except for the oxyanion hole region, Glu53 and the carbonyl oxygen atom on the tail amide bond formed a proportion of hydrogen bonds in complex 1 and complex 2. This hydrogen bonding can hold the tails of the inhibitors to a certain extent, which may explain the lighter swing of the tails of inhibitor 1 and inhibitor 2. The above hydrogen bonding rates were generally compatible with the results of the conformational observations.

### 2.8. Secondary Structure Analysis

Section 2.6 conformational insights revealed the disruption of the α4 helix underlying the lid domain in complex 1. The secondary structure analysis was carried out using the do_dssp tool in order to confirm that the variation was not an isolated phenomenon in the sample. Changes of the whole secondary structure species in the four complexes and independent protein with the simulation time are shown in Appendix A. In five molecular systems, the regions with relatively large variations in the secondary structure were in the lid domain. Figure 10 shows the secondary structure changes of the lid domain in the four complexes and the independent protein. It was found that the α6 helix was almost unchanged, while the α5 helix, α4 helix, and helix_180–184_ all showed more prominent variations. The α5 helix was unwound in a larger portion of independent protein, complex 1 and complex 4. This change resulted in increased flexibility in the region, which was consistent with the RMSF results. In complex 1, it could be seen that the α5 helix was unwound into a “bend” in the Lys188-Glu190 segment and a “turn” in the Val191–Asp192 segment after 90 ns of the simulation. In complex 2, a smaller proportion of α5 helix was unwound, but it was transformed into a “bend” in the Lys188-Thr189 segment at the end of the simulation. The α5 helix in complex 3 had the smallest percentage of unwinding and it was dominated by the “helix” in complex 3. In complex 4, the Lys188–Thr189 segment of the α5 was unwound into a “bend” after 40 ns, and the proportion of the “turn” gradually increased over 160 ns. The α5 helix in the independent protein has a large proportion of “turn” or “bend” in the Lys188–Thr189 segment for almost the entire simulation with the rest being unwound as a “bend” in some proportion. The C-terminal part of the α5 helix in all five simulated systems was partially unwound into “turn”. This variation may be related to the changes in the position of the inhibitor tail. The helix_180–184_ mainly fluctuated between helix and turn, with a greater proportion of helix at the start of the simulation and a greater proportion of turn at the end of the simulation in independent protein, which was opposed in complex 1. The helix_180–184_ in complex 2, complex 3, and complex 4 contained a lower proportion of helix that was more equally distributed. A portion of the α4 helix in the four complexes became “turn”, with a larger proportion of “turn” in complex 1 and complex 4, but less variation in the other two complexes and independent protein. This change may be related to the different polarity of the substituted groups on the head of the aromatic ring closer to the helix in the inhibitor. The above variations in the secondary structure were likely to induce changes in the binding patterns between protein and inhibitors, which would affect the inhibitory capacity of the competitive inhibitor.

### 2.9. Binding Free Energy Calculation

The binding free energy provided information on the interaction between protein and ligand, in particular the binding free energy of decomposition, which could be used to quantify the interaction of individual residues with the protein for identifying the key residues bound to the ligand. In the present work, the binding free energies of four complex systems were calculated using the MM/PBSA method.

The various binding free energy contributions are listed in Table 3 for the protein–ligand interactions of the four complex systems, with the total binding free energy (ΔG_binding_) being −180.067, −154.123, −146.621, and −75.383 kJ·mol^−1^ in that order. In all complexes, the van der Waals contribution (ΔE_vdW_) was greater than the electrostatic contribution (ΔE_elec_), indicating that van der Waals interactions were the main contributions to the protein–ligand binding. Both the van der Waals and electrostatic contributions of complex 4 were apparently less than those of the first three groups, which may be related to the fact that inhibitor 4 had a larger tail swing and the residues that interacted with it were more different from the first three inhibitors.

The decomposed binding free energies of the residues for the four systems were then calculated as listed in Appendix A, with the larger contributing residues and their binding free energies shown in Figure 11. As can be seen from the figure, Gly50, Ala51, Ser122, and Met123 exhibited a high E_MM_ contribution (understood as the sum of electrostatic and van der Waals contributions) in the four complexes, and these residues were all distributed near the oxyanion hole, indicating a strong interaction with the carbonyl oxygen of the head in this region. Leu241 near the aromatic ring of the inhibitor head exhibited more E_MM_ contribution in the four complexes as well. This residue had a large hydrophobic side chain, which together with another hydrophobic residue, Leu213, with a smaller binding free energy, sandwiched the aromatic ring of the head from both sides and formed a sandwich structure, resulting in strong binding of the head.

The residue with the most E_MM_ contribution near the benzene ring of the inhibitor tail was Tyr194, the benzene ring on the side chain of this residue was close to the benzene ring of the inhibitor tail and almost paralleled it, with a strong π–π stacking interaction between the two. In the different complexes, the inhibitor tail experienced a large swing and Tyr194 moved with it. This residue was located on the α5 helix which was the main influence on the conformational changes found in the conformational observations of the α5 helix. Another residue with a higher E_MM_ contribution to the inhibitor tail benzene ring was Val270 which had a larger hydrophobic side chain and, together with Tyr194, sandwiched the tail benzene ring, forming a sandwich structure as well.

The above residues have been mentioned in previous reports in the literature; however, since those works only performed docking studies, some of the residues they referred to that may have strong interactions, such as Leu148 and Val191, did not show a strong binding free energy contribution in our work. In addition, all the above interactions are significantly reduced in complex 4, which was not revealed by docking.

Glu53 had the strongest E_MM_ contribution in the first three complexes, while the residue had a positive E_MM_ contribution in complex 4, reflecting a repulsive interaction. Conformational observation showed that the carboxyl group on the side chain of this residue was relatively close to the -NH-group on the amide bond at the tail of the inhibitor. Although hydrogen bonding analysis has indicated that in most snapshots the distance between them exceeded the hydrogen bond range, both the hydrogen atoms and the carbon atoms on the amide bond were more positively charged, so it was possible that there was a strong ionic interaction with the negatively charged carboxyl group on the Glu53. In complex 4, due to the flipping of the inhibitor tail, the positively charged atoms in the amide bond turn to the side away from the carboxyl group on Glu53, while the negatively charged oxygen atoms shift to the side close to this carboxyl group, thus transforming from electrostatic attraction to electrostatic repulsion between the two, resulting in an opposite E_MM_.

Similar to Glu53, the three residues with charged side chains, Asp197, Asp239, and His269, can associate ionic interactions with charged groups on the amide bond of the inhibitor tail. The E_MM_ contribution of His269 in the four complexes reflected the order of 1, 4, 2, and 3, while Asp239 was in the opposite order. Asp197 roughly reflected a similar order to His269. In complex 1, Asp197 was primarily in close proximity to the positively charged -NH-group on the amide bond and formed ionic interactions. The other two residues formed an Asp239...His269...O=C chain ion interaction with the negatively charged C=O group on the other side of the amide bond. The position of Glu53 related to the three residues is shown in Figure 12. Taking the C-N bond at the inhibitor tail amide bond as the axis by placing it vertically on the paper, Glu53 and Asp197 were in the 8 and 6 o’clock directions, respectively. His269 was in the 12 o’clock direction. Asp239 was between 12 and 1 o’clock. The oxygen atoms on the amide bond were directed in the 1-point, 3-point, 4-point, and 11-point directions in complexes 1–4. With respect to Glu53, complexes 1–3 were all -NH-closer to it, reflecting electrostatic attraction, especially the -NH-groups of complex 1 and complex 2, which were closer to Glu53 with stronger gravitational forces, while complex 4 was C=O closer to it, showing repulsion. For Asp197, the -NH-group was closest to it in complex 1, similarly distant in complex 2 and complex 4, and furthest away in complex 3, with the gravitational force gradually decreasing and the repulsive force of C=O gradually increasing on the other side. The distance to C=O in the amide bond increased in the order of 1, 4, 2, and 3 for His269, with the gravitational force becoming gradually smaller and finally changing to a repulsive force in complex 3. Asp239, which lay behind His236, was the opposite of this. The largest variation in the binding free energy of the inhibitor tail amide bond to its nearby residues can be seen in Figure 10 (the purple region in the figure). Therefore, the different directions of the inhibitor tail amide bond were the main reason for the difference in affinity between the protein and the inhibitor.

Another significant difference between the different complexes was Lys273, which interacted appreciably more with the inhibitor in complex 3 and complex 4 than in complex 1 and complex 2, especially in complex 1, and even demonstrated repulsive forces. We have found that this is related to the direction of the Cl atom on the benzene ring of the inhibitor tail. The inhibitor Mulliken charge revealed that both the tail benzene ring and the -NH- group of the slightly more distant amide bond had a minority positive charge, which was repulsive to the positively charged amino group of the lysine side chain, but the Cl atom on the benzene ring could form hydrophobic interactions with the hydrocarbyl part of the side chain of this residue, which behaved as a gravitational force. In complex 1 and complex 2, the Cl atom was further away from the hydrocarbyl part of Lys273, so the repulsive force was greater than or close to the gravitational force. In contrast, in complex 3 and complex 4, the Cl atom was closer to the hydrocarbyl part of Lys273 due to the flipping of the benzene ring and the gravitational force was greater than the repulsive force. Similarly, Glu190 exhibited the opposite interaction in complex 3 and complex 4 compared to that in complex 1 and complex 2.

### 2.10. The Mechanism of the Difference in the Inhibitory Capacity of Different Inhibitors

The reports of the experiments pointed out that the inhibitors studied in this work, all of which were competitive inhibitors of MAGL, showed a positive correlation between the strength of their inhibitory capacity and the affinity of the enzyme inhibitor. The previous binding free energy calculations showed that the total binding free energy ΔG_binding_ of complexes 1–4 exhibited the same order as the inhibition capacity. This change in total binding free energy, as can be seen with reference to the conformational observations and the decomposed binding free energy of the residues, was caused mainly by the flipping of the inhibitor tail. The flipping of the inhibitor tail led to some changes in the ionic interactions with the region near the tail, especially near the amide bond. These changes would cause the proteins in some complexes to be less anchored to the inhibitor tail and to swing laterally and closer towards helix_180–184_ by the action of residues on the originally more distant lid domain. The conformational change in the tail of the inhibitor, in turn, implicated the head carbonyl oxygen, which weakened the interaction near the oxyanion hole, ultimately resulting in a dramatic reduction in the overall binding free energy of complex 4, which had the largest change in the tail, and a reduction in inhibition.

Interestingly, there was no remarkable change in the position of the 1-substituted groups at the head of the inhibitor and the piperidine parent nucleus in the four complexes, but the different 1-substituted groups resulted in variable changes in the tail benzene ring. This implied that conformational changes were not passed from the inhibitor molecule itself, but rather due to different 1-substituted groups affecting the nearby protein, leading to changes in protein conformation that further caused the flipping and lateral swinging of the tail.

In order to investigate the reasons for the tail flipping of the inhibitor, the simulation of complex 4 was observed frame by frame, and it was found that the tail flipping mainly occurred between 32 ns and 34 ns, which were selected for comparative analysis against the conformation of the two frames.

The head aromatic ring of complex 4 was p-chlorophenyl, with its Cl atom exactly close to the Thr157-Lys160 region on the α4 helix. Except for Phe159, the residues in this region were all hydrophilic, which was somewhat perturbed by the proximity of the hydrophobic chlorinated phenyl group. In the 34 ns conformation, this perturbation induced a partial unwinding of the α4 helix. As a result, there was a larger conformational change for Phe159 in 34 ns compared to 32 ns, where it would move closer to the Cl atom and form a stronger hydrophobic interaction with it. This interaction could lead to a more appreciable movement of the benzene ring towards the lid domain and close to Ile179 on it, forming a new hydrophobic interaction. The hydrogen atom of the peptide plane in Ile179 originally formed a hydrogen bond with the oxygen atom of the peptide plane of Arg202, anchoring the side chain of Arg202 towards the solvent and away from Tyr194 on the other side. The conformational change of Ile179 resulted in the disruption of the hydrogen bond between it and Arg202, which allowed the side chain of Arg202 to undergo a large swing in the solvent. As it approached Tyr194, its side chain guanidyl group would form a new hydrogen bond with the oxygen in the peptide plane of Tyr194 and flip the side chain benzene ring of Tyr194. As discussed in Section 2.9, the Tyr194 interacted strongly with the inhibitor tail benzene ring via π–π stacking, and its flipping caused the inhibitor tail benzene ring to follow the flip. The whole process is shown in Figure 13. The secondary structure analysis in Section 2.8 has shown that the perturbation of the α4 helix differed when the type of 1-substituted group was different, so it caused a different flipping of the inhibitor tail.

## 3. Calculation Methods

### 3.1. Acquisition of Monoacylglycerol Lipase and Inhibitor Structures

The X-ray crystal structure of human MAGL (PDB ID: 3PE6, resolution: 1.35 Å) was obtained directly from the RSCB database [35] (https://www.rcsb.org (accessed on 24 December 2021)) according to the literature [32]. The 3D structure of MAGL was obtained by manually removing the crystalline water molecules and the compounds that assisted in the crystallization of the protein. The four small molecular inhibitors were constructed in Gauss view 6.0.16 and optimized with the Gaussian16 package under the B3LYP/6-31G(d) motif [36], eventually yielding reasonable small molecular structures.

### 3.2. Molecular Docking

The AutoDock Vina 1.1.2 software (It was originally designed and implemented by Dr. Oleg Trott in the Molecular Graphics Lab (now CCSB) at The Scripps Research Institute) was used to dock proteins and inhibitors in this study [37,38,39]. Based on the binding sites of reversible inhibitors of monoacylglycerol lipase reported in the literature [21], a docking box was positioned around the catalytic triad. The docking box was set to a size of 16 Å × 14 Å × 26 Å with a 1 Å grid space. The AutoDock Vina program was then searched for conformations to score the binding patterns of the protein–ligand complexes. The conformations that were highly scored and corresponded to the binding patterns recorded in the literature [32] were chosen as the initial conformations for the molecular dynamics simulations.

### 3.3. Molecular Dynamics Simulations

A classical dynamics simulation of the above initial conformations was conducted using the GROMACS 4.5.2 package [40]. The protein parameter file was constructed by applying the AMBER99SB-ILDN force field [41]. The GAFF [42] (General AMBER Force Field) force field parameter file was created for the ligand with the antechamber program of the AMBER16 package [43]. The entire system was solvated using the SPC water model [44] in which 1 sodium ion was added to balance the charge. The next step is to minimize the energy of the system in the steepest descent algorithm. Two stages of equilibration were then implemented. The first stage was an NVT equilibration of 500 ps at 300 K using the V-rescale coupling algorithm. The second stage was an NPT equilibration of 500 ps at the same temperature using the Parrinello-Rahman coupling algorithm with a pressure setting of 1 atm. Eventually, molecular dynamics simulation was performed for 200 ns in an equilibrium environment, and the LINCS algorithm was used to constrain all bonds during the simulation. The long-range electrostatic force was calculated by the particle mesh Ewald algorithm with a grid spacing of 1.6 Å. Furthermore, the time integration step was 2 fs, and the position coordinates and energy were saved every 10 ps. The modelling trajectory was finally obtained for the next step of the analysis.

### 3.4. Binding Free Energy Calculation

The MM/PBSA method is an efficient way to compute the binding free energy for different complex systems [45,46]. There were 1000 frames of conformation selected from the equilibrium trajectories of molecular dynamics simulation with a duration of 20 ns, and the protein–ligand complexes were subjected to binding free energy calculations using the g_mmpbsa tool in our work. The binding free energy is defined as follows:∆G_bind_ = G_complex_ − (G_protein_ + G_ligand_),
where G_complex_ indicates the total free energy of the complex. G_protein_ and G_ligand_ are the total free energies of protein and ligand separated in the solvent, respectively. However, each of the three individual entity free energies of the complex, the protein, and the ligand can be further described as the following:G = <E_MM_> − TS + <G_solv_>.

The free energy G is divided into two terms: the molecular mechanics term and the solvation energy. <E_MM_> indicates the average molecular mechanics potential energy in a vacuum, <G_solv_> indicates the solvation free energy, and TS refers to the contribution of entropy to the free energy in a vacuum.

The vacuum molecular mechanics potential energy (E_MM_) is the sum of the energies of bonded (E_bonded_) and unbonded (E_nonbonded_) interactions. Here, E_bond_, E_angle_, and E_torsion_ are denoted as bond, bond angle, and dihedral angle interactions, respectively, while E_elec_ and E_vdW_ are denoted as Coulomb and van der Waals interactions, respectively, and are modelled using the Coulomb and Lennard–Jones potential functions, respectively. The free energy of solvation (G_solv_) is divided into two parts, G_polar_ and G_nonpolar_, with G_polar_ and G_nonpolar_ representing the free energies of polar and non-polar solvation, respectively. The above equation can be indicated in the following way:E_MM_ = E_bonded_ + E_nonbonded_;
E_bonded_ = E_bond_ + E_angle_ + E_torsion_;
E_nonbonded_ = E_elec_ + E_vdW_;
G_solv_ = G_polar_ + G_nonpolar_.

## 4. Conclusions

Our work focused on the differences in the enzyme–inhibitor binding patterns for different 1-substituted groups of the aryl formyl piperidine parent nucleus and the reasons for their different inhibitory effects in order to reveal the inhibitory mechanism of the inhibitors at the molecular level.

The results of the study showed that in the four systems, the regions where the enzyme interacted strongly with the inhibitor included near the head aromatic ring, near the head carbonyl oxygen, near the tail amide bond, and near the tail benzene ring of inhibitors. The binding free energy near the tail amide bond varied considerably between the different complexes, and the interactions in the other three regions of complex 4 were significantly reduced.

Several analyses revealed that differences in the 1-substituted group produced a more apparent conformational change of the more flexible lid domain in the enzyme. This variation could cause a different degree of flipping in the tail of the inhibitor. The different direction of the tail amide bond resulting from the flipping caused a large change in the interaction with some residues carrying charged side chains in the enzyme and further contributed to a lateral swing of the tail benzene ring towards helix_180–184_ in complex 3 and complex 4. Among them, complex 4 additionally weakened the binding strength between the head carbonyl oxygen and the oxyanion hole due to the stronger swing, leading to the reduction of its overall binding free energy. The above findings implied important implications for the purposeful design of aryl formyl piperidine derivative-based inhibitors of MAGL.

## Figures and Tables

**Figure 1 molecules-27-07512-f001:**
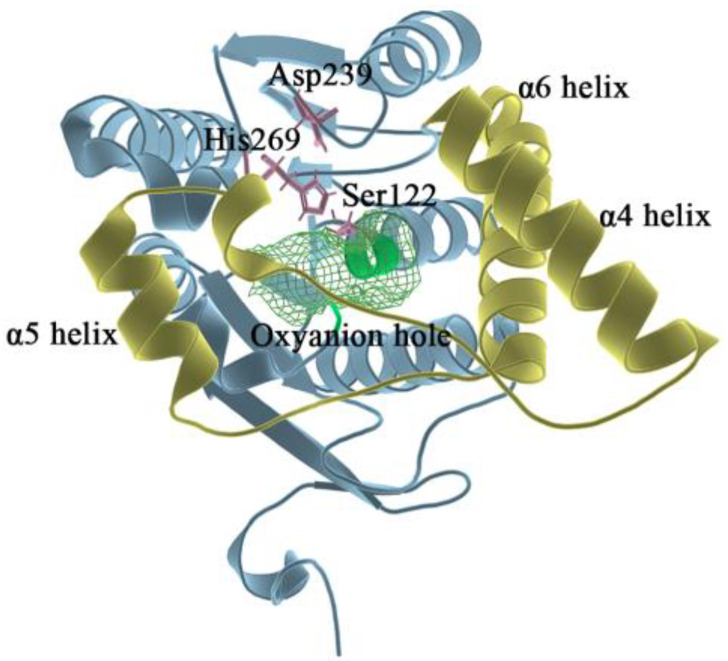
In the overall 3D structure of monoacylglycerol lipase, yellow represents the lid domain, pink represents the catalytic triads, green mesh pattern represents the oxyanion hole, and other regions are shown in blue.

**Figure 2 molecules-27-07512-f002:**
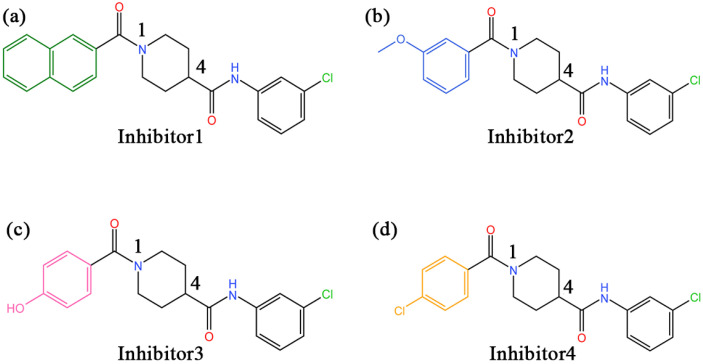
(**a**–**d**) The 2D structures of inhibitors 1–4 and the groups in each inhibitor, from left to right, are referred to as the head aromatic ring, the head carbonyl group, the piperidine parent nucleus, the tail amide bond, and the tail benzene ring, respectively, in this paper.

**Figure 3 molecules-27-07512-f003:**
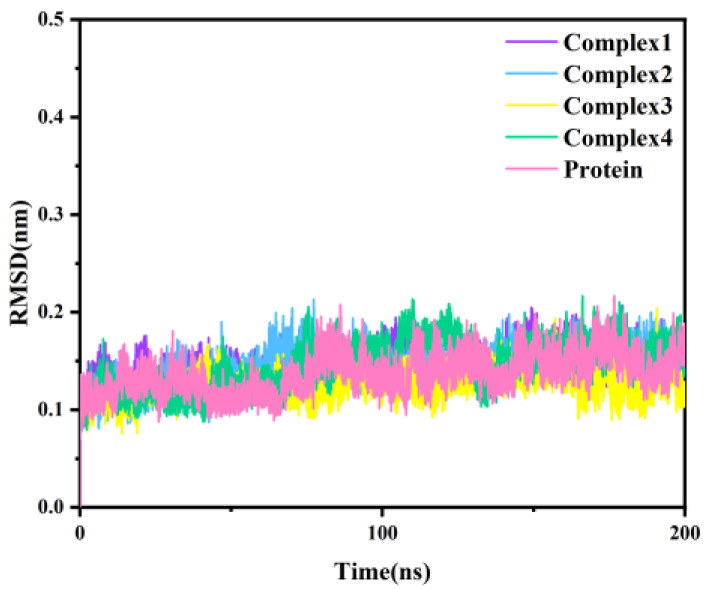
The RMSD of the Cα atomic backbone of the four complexes and independent protein with simulation time.

**Figure 4 molecules-27-07512-f004:**
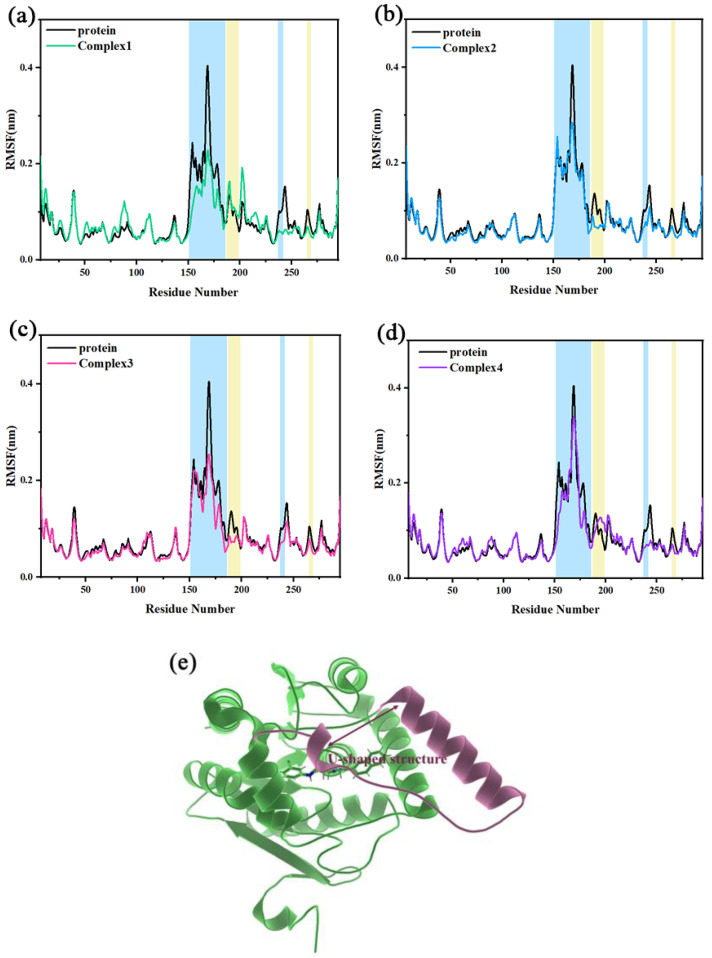
(**a**–**d**) The RMSF of independent protein and the four complexes are compared, with several areas of significant difference highlighted in blue and yellow backgrounds, respectively. (**e**) The position of the U-shaped structure in the protein.

**Figure 5 molecules-27-07512-f005:**
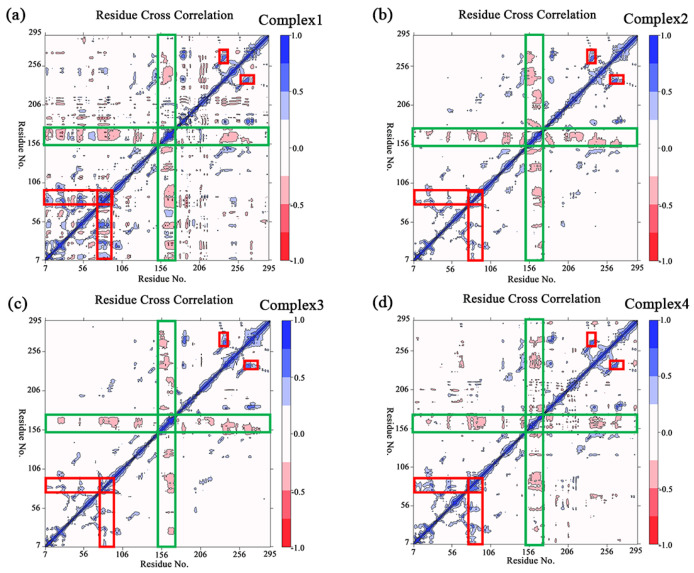
(**a**–**d**) The plot of the covariance matrix for the complexes 1–4, the red–to–blue bar legend represents the degrees of correlation between residues, where blue indicates a positive correlation and red indicates a negative correlation. Several significant areas of correlated motion are highlighted with rectangles, of which the negative correlation is circled in green, and the positive correlation is circled in red.

**Figure 6 molecules-27-07512-f006:**
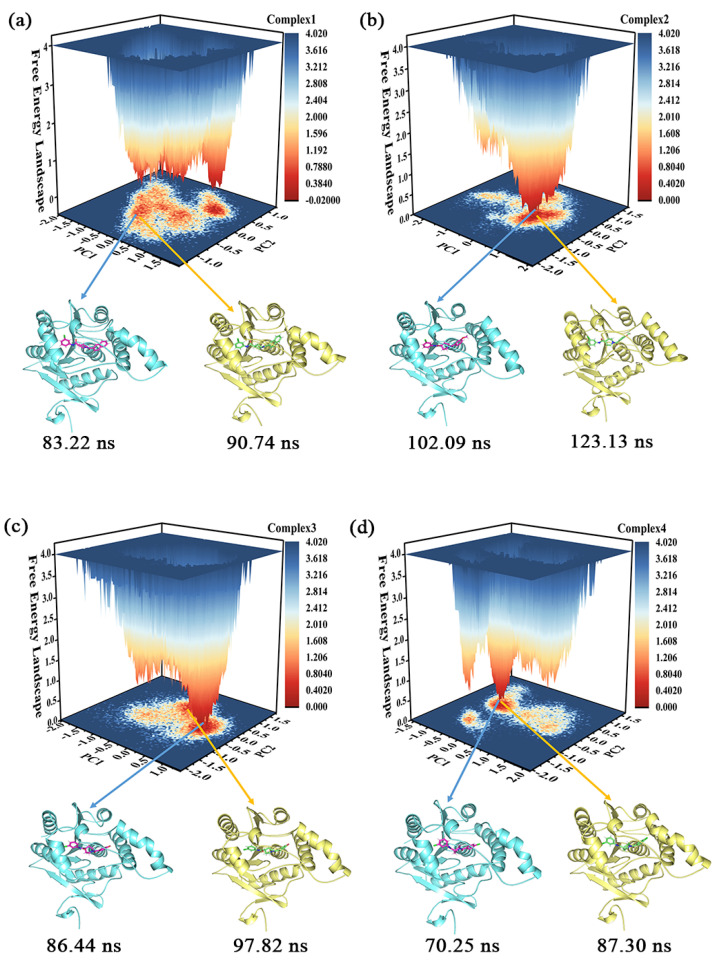
(**a**–**d**) The free energy landscape of the complexes 1–4 and their representative conformations were extracted from the MD simulation trajectories.

**Figure 7 molecules-27-07512-f007:**
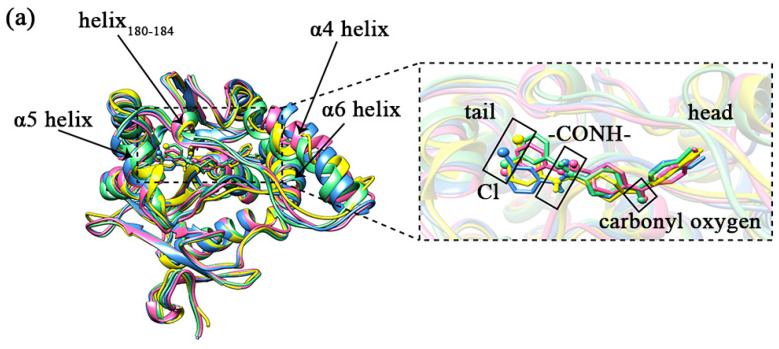
(**a**) The conformations of the four complexes are superimposed. The MAGL is shown in ribbon mode. The inhibitor is shown in stick mode, where the carbonyl oxygen at the head, the amide bond at the tail, and the Cl atom on the tail benzene ring are specifically shown in stick and ball mode. Complex 1 is shown in yellow, complex 2 is shown in blue, complex 3 is shown in pink, and complex 4 is shown in green. (**b**–**e**) A comparative diagram showing the superposition of α4 helix, helix_180–184_, α5 helix, and α6 helix in the lid domain of the four complexes; yellow represents complex 1, blue represents complex 2, pink represents complex 3, and green represents complex 4.

**Figure 8 molecules-27-07512-f008:**
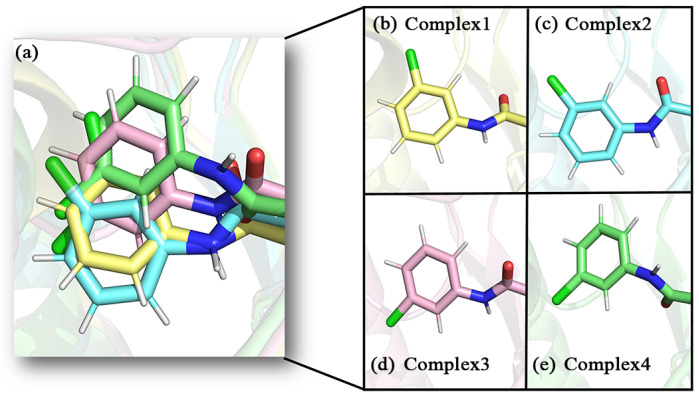
Enlarged detail showing the direction of the tail benzene ring in the four complexes, (**a**) superimposed diagram of the four complexes, (**b**–**e**) detail of the four complexes in the benzene ring direction.

**Figure 9 molecules-27-07512-f009:**
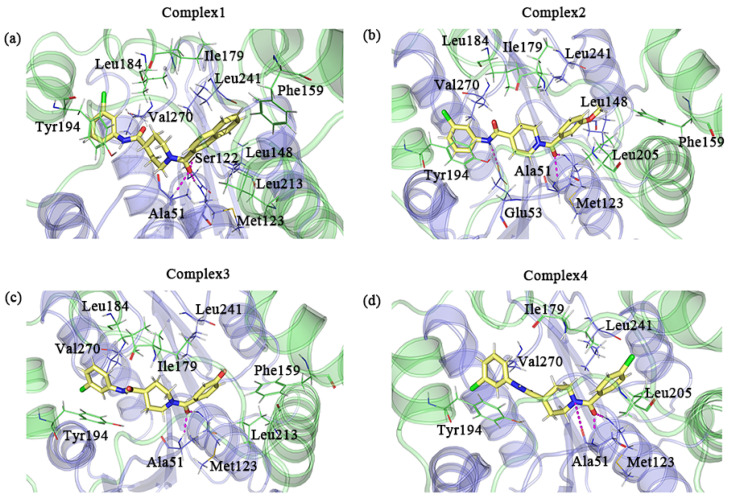
(**a**–**d**) A detailed diagram of the conformational observations for the complexes 1–4; the protein and residues of the lid domain are shown in green, and the rest of the protein and residues are shown in blue–purple. The key residues in complexes 1–4 that interact with the inhibitor are indicated by lines, where the inhibitor is indicated by the yellow stick mode. Hydrogen bonds are shown as pink dashed lines.

**Figure 10 molecules-27-07512-f010:**
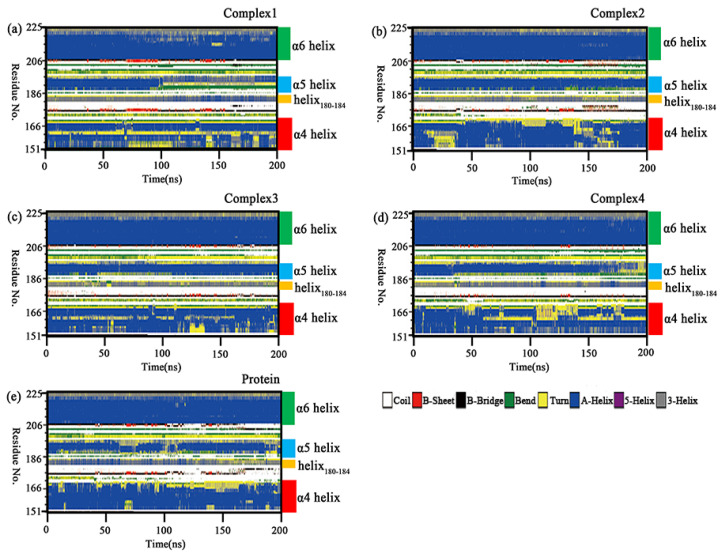
(**a**–**e**) Changes in the secondary structure of the lid domain (Ala151–Pro225) in the complexes 1–4 and independent protein with simulation time.

**Figure 11 molecules-27-07512-f011:**
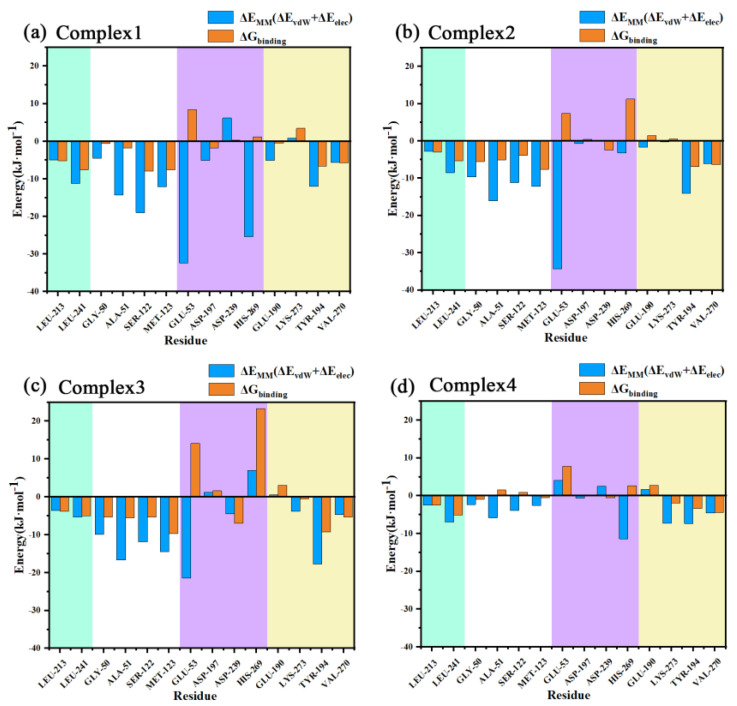
(**a**–**d**) Decomposition energies of the important residues in the complexes 1–4 are shown from left to right in each figure for residues near the head benzene ring (green areas), near the head carbonyl oxygen (white areas), near the tail amide bond (purple areas), and near the tail benzene ring (yellow areas).

**Figure 12 molecules-27-07512-f012:**
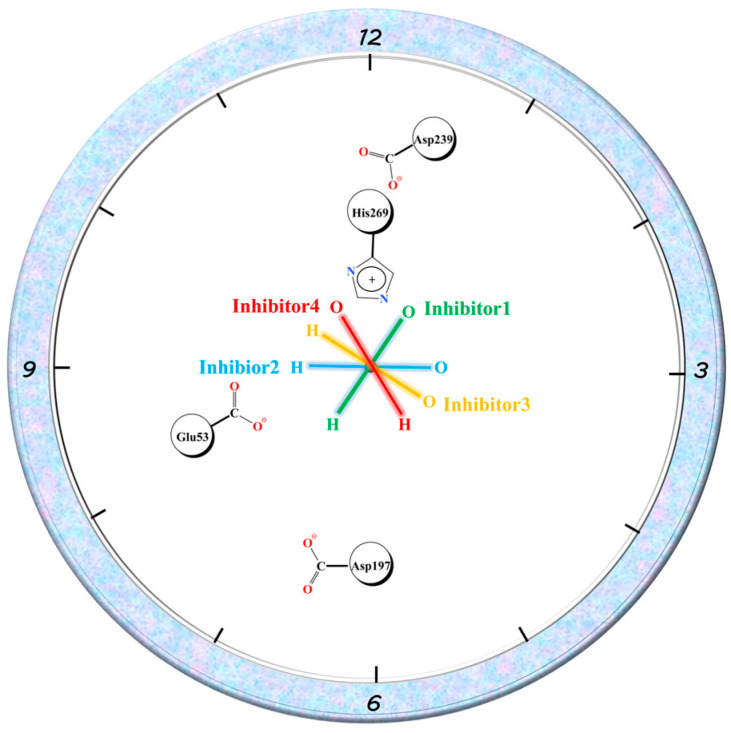
Schematic diagram of Glu53, Asp197, Asp239, and His269 residue positions and amide bond directions, where the C-N bond in the amide bond is used as the axis and is oriented perpendicular to the paper plane.

**Figure 13 molecules-27-07512-f013:**
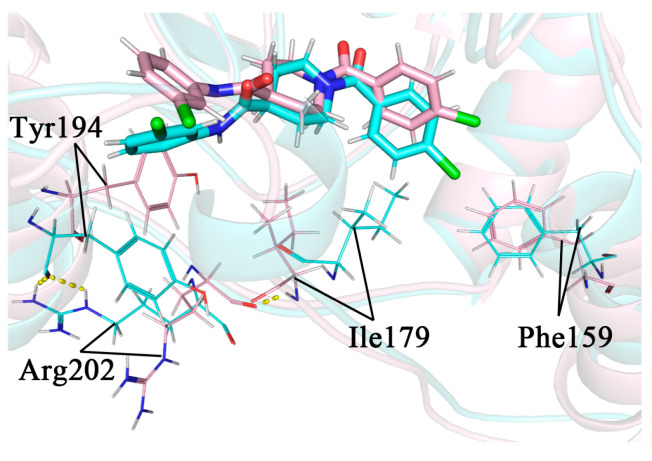
Comparison images of the superimposed snapshots of 32 ns (pink) and 34 ns (blue) in complex 4, with the flipped key residue shown in line mode and the hydrogen bond shown in yellow dashed mode. The inhibitor tail is shown on the left and the inhibitor head is shown on the right in the figure.

**Table 1 molecules-27-07512-t001:** The average values of RMSD for α4 helix, helix_180–184_, α5 helix, and α6 helix in the four complexes are calculated (nm).

Average Term of RMSD	α4 Helix	Helix_180–184_	α5 Helix	α6 Helix
Complex 1	0.1353	0.0492	0.0720	0.0827
Complex 2	0.1501	0.0327	0.0505	0.0539
Complex 3	0.1242	0.0336	0.0519	0.0581
Complex 4	0.1430	0.0379	0.0646	0.0620

**Table 2 molecules-27-07512-t002:** The occurrence percentage of major hydrogen bonds in the four complexes during the simulation.

Systems	Residues	Occurrence
Complex 1	Ala51	27.36%
	Glu53	14.13%
	Ser122	53.84%
	Met123	70.39%
Complex 2	Ala51	58.77%
	Glu53	42.09%
	Met123	71.84%
Complex 3	Ala51	71.83%
	Met123	79.50%
Complex 4	Ala51	29.85%
	Met123	22.30%

**Table 3 molecules-27-07512-t003:** Different energy components and binding free energies of the four complexes were calculated by MM/PBSA (kJ·mol^−1^).

Energy Contribution	Complex 2	Complex 3	Complex 3	Complex 4
ΔE_vdW_	−211.150	−207.610	−208.967	−136.215
ΔE_elec_	−145.935	−103.245	−103.559	−36.980
ΔG_PB_	196.135	176.906	186.153	112.724
ΔG_SA_	−19.276	−20.124	−20.243	−14.657
ΔG_polar_ ^a^	50.200	73.661	82.594	75.744
ΔG_nonpolar_ ^b^	−230.426	−227.734	−229.210	−150.872
ΔG_binding_ ^c^	−180.067	−154.123	−146.621	−75.383

(^a^ ΔG_polar_ = ΔG_elec_ + ΔG_PB_, ^b^ ΔG_nonpolar_ = ΔG_vdW_ + ΔG_SA_, ^c^ ΔG_binding_ = ΔG_polar_ + ΔG_nonpolar_).

## Data Availability

Not applicable.

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
