# Peer review of "Insight into the Inhibitory Mechanism of Aryl Formyl Piperidine Derivatives on Monoacylglycerol Lipase through Molecular Dynamics Simulations"

_molecules, 2022, doi:10.3390/molecules27217512_

Round 1

Reviewer 1 Report

An interesting and detailed in silico study dedicated to the subtle differences between the inhibitory effects of four structurally distinctive reversible inhibitors of monoacylglycerol lipase.

I have only two observations, the first regarding a possible misuse of an acronym, and the second regarding the intelligibility of a Figure.

R1. Based on the context of the phrase, is it possible that in line 40 not ECS but CNS should be the appropriate acronym?

Perhaps the meaning of your assertion is: "Endocannabinoid system regulates the activity of the CNS, which may have therapeutic potential ...".

R2. In Figure 8, could you distinctively color the range between ALA151 and Pro225 residues? It will be easier for the readers to understand the complex conformation and the conformational changes of the regulatory lid-domain of MAGL, which encompasses the involved inhibitors. (NOT MANDATORY)

Reviewer 2 Report

The manuscript by Liu, et al. titled “Insight into the inhibitory mechanism of Aryl Formyl Piperidine derivatives on Monoacylglycerol Lipase through Molecular Dynamics Simulations” investigate the inhibitory mechanism and binding modes of four aryl formyl piperidine derivatives inhibitors with different 1- substituents to monoacylglycerol lipase (MAGL). The difference in 1-substituted group caused significant conformational changes in the lid domain of the enzyme.

Overall, the manuscript is fascinating and has a valuable contribution to the scientific field and society. I recommend this manuscript be published at Molecules after some minor corrections:

1.    The chemical name of the inhibitors should be written in the manuscript. Currently, they are only written as Inhibitor 1 – 4.

2.    In the methods part, authors should specify how the compounds and the protein structure were prepared for molecular docking and MD simulations.

3.    In lines 124 – 129, the authors discussed the results of the molecular docking of the inhibitors to the protein structure. They discuss the hydrogen bond interactions between the ligands and the proteins. I suggest authors provide a figure showing these interactions as it is not clear in the current form.

4.    In line 139, authors wrote (pointing to Figure 3): “…all five systems were balanced at approximately 20 ns after an initial rapid rise.” However, it is not quite clear where the rapid rise after the 20 ns position is in Figure 3. I suggest the authors clary the statement (e.g. by adding an arrow in the figure) as I saw the RMSDs have been stable since the beginning.

5.    In line 149, the authors discussed the RMSF values of the U-shaped structure  (Ala151-Ser176) for the four complexes that became smaller. First, the authors need to clarify what the U-shaped structure means. Second, the authors need to write how small the RMSF values are, probably by putting the number in the bracket. Currently, the statement tends to be qualitative. Third, the authors need to add a figure showing the position of the U-shaped structure in the protein structure to clarify the statement.

6.    In Figure 5, the authors should explain the meaning of the red and green rectangles and the red-to-blue bar.

7.    In lines 208-209, the authors should mention the RMSD differences among the lid domains and show which one is the highest and the lowest.

8.    In line 214, the authors mentioned that a4 helix displayed the greatest conformational changes (pointing to Figure 7). However, Figure 7 is not clearly depicting this claim. I suggest the author add an inset to the figure to show the claim and also add the RMSD values to the text. The same clarifications should also be made for the following sentences (lines 215-220).

9.    In lines 230-231, the authors need to add a figure showing the flipping of the tail benzene ring as currently not clearly depicted in Figure 7.

10. In line 259, the significant numbers are too many and should be fixed.

11. In line 260, the authors should clarify the correlation between the number of hydrogen bonds to the IC50 values of the inhibitors.

12. In lines 289-290, the authors stated, “The protein was stabilized in "turn" at the end of the  simulation, while both complex 1 and complex 4 were stabilized in "bend" and "turn".” I found the term “turn” and “bend” confusing and not clearly depicted in Figure 9. Please clarify this.
